# Gut Microbiota and Obsessive–Compulsive Disorder: A Systematic Review of Mechanistic Links, Evidence from Human and Preclinical Studies, and Therapeutic Prospects

**DOI:** 10.3390/life15101585

**Published:** 2025-10-10

**Authors:** Shayan Eghdami, Mahdieh Saeidi, Sasidhar Gunturu, Mahsa Boroon, Mohammadreza Shalbafan

**Affiliations:** 1Brain and Cognition Clinic, Institute for Cognitive Sciences Studies, Tehran 01449614535, Iran; seghdami33@gmail.com; 2Department of Psychiatry, BrnoxCare Health System, New York, NY 10456, USA; msaeidi@bronxcare.org (M.S.); sguntur1@bronxcare.org (S.G.); 3Department of Psychiatry, Icahn School of Medicine at Mount Sinai, New York, NY 10029, USA; 4Department of Psychiatry, Imam Hossein Hospital, School of Medicine, Alborz University of Medical Sciences, Karaj 3149779453, Iran; mahsa.boroon@gmail.com; 5Mental Health Research Center, Psychosocial Health Research Institute (PHRI), Department of Psychiatry, School of Medicine, Iran University of Medical Sciences, Tehran 01449614535, Iran

**Keywords:** OCD, gut microbiota, gut-brain axis, fecal microbiota transplantation (FMT), intestinal permeability, inflammation, biomarkers, epigenetics

## Abstract

Obsessive–compulsive disorder (OCD) is a multifactorial condition, and interest in gut–brain interactions is increasing. We conducted a systematic two-step review, registered in PROSPERO (CRD420251083936). Step 1 mapped core OCD biology to gut-relevant pathways, including neuroimmune activation, epithelial barrier function, microbial metabolites, and stress circuitry, to clarify plausible mechanisms. Step 2 synthesized evidence from human and preclinical studies that measured or manipulated microbiota. Searches across PubMed, EMBASE, Web of Science, PsycINFO, and Cochrane (September 2025) yielded 357 biological and 20 microbiota-focused studies. Risk of bias was assessed using the Joanna Briggs Institute checklist for human studies and SYRCLE’s tool for animal studies. Although taxonomic findings in human cohorts were heterogeneous, functional patterns converged: reduced short-chain fatty acid capacity, enrichment of pro-inflammatory pathways, and host markers of barrier disruption and inflammation correlating with OCD severity. Transferring patient microbiota to mice induced OCD-like behaviors with neuroinflammatory changes, partly rescued by metabolites or barrier-supporting strains. Mendelian randomization suggested possible causal contributions at higher taxonomic levels. Diet, especially fiber intake, and psychotropic exposure were major sources of heterogeneity. Evidence supports the microbiota as a modifiable co-factor in a subset of OCD, motivating diet-controlled, stratified clinical trials with composite host–microbe endpoints.

## 1. Introduction

Obsessive–compulsive disorder (OCD) is a chronic, disabling neuropsychiatric condition defined by obsessions (intrusive, irrational thoughts, urges, or images) and compulsions (repetitive behaviors or mental acts that temporarily relieve distress). With a lifetime prevalence of 0.8–3% and ranking among the World Health Organization’s ten most disabling conditions worldwide, OCD represents a significant global health burden [1,2,3,4,5].

The etiology of OCD is multifactorial, reflecting complex interactions between genetic vulnerability, cortico-striato-thalamo-cortical (CSTC) circuit dysfunction, and immune, metabolic, neuroendocrine, and environmental factors [6,7,8,9].

In recent years, growing attention has focused on the gut microbiota as a potential contributor to brain and behavior via the bidirectional brain–gut–microbiome (MGB) axis [10]. The gut–brain axis operates through multiple interconnected pathways that directly relate to established OCD pathophysiology. Mechanistically, gut microbiota produce short-chain fatty acids (SCFAs), particularly butyrate, propionate, and acetate, which maintain intestinal barrier integrity and possess potent anti-neuroinflammatory properties [11,12]. Dysbiosis can compromise this barrier function, leading to increased intestinal permeability and translocation of bacterial lipopolysaccharides (LPSs), triggering systemic inflammation that can reach the central nervous system through compromised blood–brain barrier integrity [13,14].

Additionally, specific bacterial strains synthesize or metabolize key neurotransmitters implicated in OCD. Lactobacillus species produce gamma-aminobutyric acid (GABA), while Enterococcus and Bacillus species influence dopamine pathways, and various taxa modulate serotonin metabolism through the tryptophan-kynurenine pathway [15,16]. The vagus nerve provides direct neural communication between the gut and brain, transmitting signals that can modulate the hypothalamic–pituitary–adrenal (HPA) axis and influence microglial activation [17]. These pathways converge on neural circuits implicated in OCD, including the CSTC networks, through neuroinflammatory processes, neurotransmitter imbalances, and epigenetic modifications mediated by microbial metabolites acting as histone deacetylase inhibitors [18].

The evidence supporting gut–brain axis involvement in OCD shows significant disparities between animal models and human studies. Preclinical research has provided compelling mechanistic evidence with clear causal relationships: fecal microbiota transplantation (FMT) from OCD patients to germ-free mice successfully transfers compulsive-like behaviors, while antibiotic treatment can rescue repetitive behaviors in genetic OCD models [19,20]. Animal studies have demonstrated that specific microbial metabolites can induce repetitive behaviors through dopamine D1-receptor mechanisms and neuroinflammatory pathways, and probiotic interventions consistently improve OCD-like behaviors alongside restoring SCFA production [21,22]

In contrast, human evidence remains more limited, heterogeneous, and primarily correlational. While clinical studies have identified gut microbiota alterations in OCD patients, including reduced alpha diversity and decreased butyrate-producing species, taxonomic findings show poor replication across cohorts [11,23]. The most reproducible human findings involve functional changes such as reduced SCFA biosynthetic capacity, elevated inflammatory pathways, and host biomarkers of barrier dysfunction that correlate positively with symptom severity [24,25]. Importantly, whether these microbiome alterations are causative, consequential, or correlational with OCD pathophysiology remains uncertain. Human studies are further complicated by confounding factors largely absent from animal models, including medication effects, dietary patterns, and the bidirectional nature of OCD symptoms potentially influencing microbiome composition [26].

Several biological domains implicated in OCD pathophysiology show sensitivity to microbiome perturbations, creating potential mechanistic bridges between gut dysbiosis and OCD symptomatology. The immune system represents a key convergence point: OCD patients consistently show elevated pro-inflammatory markers (IL-6, TNF-α), while dysbiotic microbiomes promote systemic inflammation through LPS translocation and reduced anti-inflammatory metabolite production [27,28]. The SCFA-mediated pathway appears particularly relevant, as butyrate deficiency compromises both intestinal barrier integrity and microglial homeostasis [29].

Neurotransmitter systems central to OCD pathophysiology also interface with microbial metabolism. Alterations in the tryptophan-kynurenine pathway can be influenced by gut microbiota through direct metabolite production and inflammatory cascade activation [30]. Similarly, GABA and glutamate balance, critical in OCD’s cortico-striatal circuitry, can be modulated by specific bacterial strains [31]. The HPA axis dysfunction common in OCD intersects with gut–brain signaling through cortisol’s effects on intestinal permeability and microbiome composition, creating potential feedback loops [32].

To address the critical gap between robust preclinical evidence and inconsistent human findings, we designed a two-step systematic review strategy that explicitly maps established OCD biological pathways onto gut microbiome mechanisms. Step 1 synthesizes evidence on biological systems consistently altered in OCD, including immune activation, barrier dysfunction, neurotransmitter imbalances, HPA axis dysregulation, oxidative stress, and epigenetic modifications. Step 2 evaluates human and animal studies that directly relate gut microbiome composition or function to these same biological systems.

This framework allows us to move beyond simple taxonomic associations toward functionally relevant mechanisms that could inform therapeutic interventions. Our synthesis explicitly addresses the animal–human evidence gap by evaluating the strength of mechanistic support across different experimental contexts, with particular attention to replicable functional signatures (SCFA deficiency, barrier dysfunction, inflammatory activation) that may represent targetable pathways for microbiome-based interventions in OCD.

## 2. Materials and Methods

### 2.1. Protocol and Registration

This review follows the Preferred Reporting Items for Systematic Reviews and Meta-Analyses (PRISMA) 2020 statement and was conducted according to a protocol registered in PROSPERO (ID: CRD420251083936) [33]. The manuscript reports findings in two steps, which address (i) biological and molecular alterations associated with OCD and (ii) studies directly linking gut microbiota composition or function to OCD.

### 2.2. Eligibility Criteria

We included experimental and clinical studies that met the following criteria:

Population: Human adults with a diagnosis of OCD based on established diagnostic criteria from the Diagnostic and Statistical Manual of Mental Disorders (DSM), published by the American Psychiatric Association, or the International Classification of Diseases (ICD), published by the World Health Organization [34,35]. And animal studies with validated OCD-like models (such as SAPAP3 knockout mice, quinpirole-induced model, natural compulsive phenotypes).

Exposure (first step): Biological, molecular, or cellular alterations in immune, metabolic, neuroendocrine, epigenetic, or neurotransmitter systems relevant to OCD, independent of microbiota measures.

Exposure (second step): Gut microbiota composition, gut dysbiosis, or interventions affecting microbiota (probiotics, antibiotics, fecal microbiota transplantation, diet) assessed in relation to OCD.

Outcomes:Immune and inflammatory markers (such as cytokines, acute-phase reactants, immune cell phenotypes);Metabolic and mitochondrial parameters (such as SCFAs, organic acids, oxidative phosphorylation activity);Neuroendocrine measures (such as cortisol, ACTH);Epigenetic modifications (DNA methylation, histone modifications, non-coding RNAs);Neurotransmitter systems (glutamate, GABA, serotonin, dopamine);Barrier function markers (such as zonulin, occludin);OCD symptom severity or compulsive-like behavior in animal models.

Studies were excluded if they met any of the following criteria:

Study type: Review articles, meta-analyses, editorials, commentaries, or opinion pieces, conference abstracts, poster presentations, or proceedings without full peer-reviewed data, case reports or case series with fewer than 10 participants, and studies without appropriate control groups (for observational studies).

Population exclusions: Studies focusing primarily on pediatric or adolescent populations (under 18 years), studies where OCD diagnosis could not be clearly isolated from other primary psychiatric conditions, studies including patients with major psychiatric comorbidities that could confound microbiome results unless OCD was clearly the primary focus, and studies of patients with significant medical comorbidities affecting gut function.

Methodological exclusions: In vitro studies including cell culture or laboratory-based studies without human or animal subjects, animal studies that did not use established OCD-like behavioral paradigms, studies where microbiota data could not be clearly isolated or distinguished from other biological measures, studies without adequate microbiome profiling methods.

Data quality exclusions: Studies with incomplete outcome data that could not be extracted or calculated, studies published in languages other than English, and duplicate publications of the same dataset.

Intervention-specific exclusions (for step 2): Studies of interventions not directly targeting or measuring gut microbiota and studies measuring only peripheral biomarkers without concurrent microbiome analysis (except for established gut–brain axis markers like zonulin/occludin).

### 2.3. Information Sources

We searched PubMed/MEDLINE, EMBASE, Web of Science, PsycINFO, and the Cochrane Library from their inception until the end of September 2025.

For the first step, search strings combined terms for OCD with terms for immune, metabolic, neuroendocrine, epigenetic, or neurotransmitter alterations. For the second step, search strings combined terms for OCD with terms for gut microbiota and related concepts. Detailed search strategies for each database are provided in Appendix A.

### 2.4. Study Selection

All retrieved records were imported into Zotero, Version 7.0 [36], and duplicates were removed.

Two reviewers (SE and MS) independently screened titles and abstracts against the eligibility criteria. Full texts were retrieved for potentially relevant studies and assessed for inclusion. Discrepancies were resolved through discussion or consultation with a third reviewer.

### 2.5. Data Extraction

We used a standardized extraction form to collect study design and setting, sample size and characteristics (human: age, sex, medication status; animal: strain, model induction), exposure details (biological domain for step 1; microbiota measure or intervention for step 2), outcomes (biomarker levels, microbial diversity, taxa, barrier measures, behavioral indices), main findings, including statistical significance and direction of effect, and mechanistic vs. observational classification.

Data were extracted independently by two reviewers; any disagreement between two authors regarding quality of studies was resolved through discussion or consultation with the third author.

### 2.6. Risk of Bias Assessment

Risk of bias was evaluated according to study type. We used the Joanna Briggs Institute (JBI) checklist in the human observational studies [37], and SYRCLE’s risk of bias tool in the animal studies [38].

Assessments were conducted independently by two reviewers (SE and MS). Any disagreement between two authors regarding quality of studies was resolved through discussion or consultation with the third author.

### 2.7. Evidence Synthesis

We conducted a narrative synthesis structured along two dimensions. First, by biological pathway implicated in OCD pathophysiology, human vs. animal studies, and observational vs. mechanistic designs.

Findings from each study were summarized using the level of statistical detail reported. Where authors provided formal metrics (α/β-diversity indices, PERMANOVA outputs, adjusted regression models), these were integrated quantitatively when possible. Where such parameters were absent (descriptive results, unadjusted associations, narrative mechanistic reports), we presented the findings narratively without additional inference.

This approach led us to (i) compare convergent and divergent signals across human observational cohorts, clinical biomarker studies, genetic analyses, and experimental animal models; (ii) highlight how microbiome-related alterations mapped onto established OCD related biological pathways (immune activation, inflammation, intestinal barrier disruption, neurotransmitter and neuropeptide signaling, metabolic regulation, and epigenetic modulation); and (iii) transparently account for differences in methodological rigor, reporting standards, and inference strength across the evidence base.

## 3. Results

### 3.1. Step 1: Biological and Molecular Alterations in OCD Pathophysiology

#### 3.1.1. Overview of Included Studies

We identified 357 studies spanning human, animal, and in vitro work (Figure 1). We have classified the domain of the biomarkers as monoaminergic (N = 127 studies), epigenetic and gene regulation (N = 92), glutamatergic/GABAergic (N = 81), immune and inflammatory (N = 82), other peripheral/metabolic (N = 33), oxidative and nitrosative (N = 29), HPA axis and neuroendocrine (N = 29), and neurotrophic and plasticity (N = 28) (Appendix A).

#### 3.1.2. Glutamatergic and GABAergic Dysfunction in OCD

Across this literature, glutamatergic markers such as glutamate, mGluR5 signaling, and GABAergic measures such as GABA and receptor or transporter indices, were repeatedly examined. Multiple studies described elevated glutamatergic signaling or higher glutamate-related readouts in OCD contexts, whereas GABAergic measures were often reported as reduced. Several datasets also assessed Glx and glycine, with glycine increases recurrently [39,40,41,42,43].

#### 3.1.3. Monoaminergic System Alterations in OCD Pathophysiology

This was the most frequently studied domain. Findings showed serotonergic, dopaminergic, and noradrenergic measures. Across studies, IL-6/CRP-adjacent oxidative/inflammatory readouts embedded within monoaminergic paradigms were commonly abnormal (see Section 3.1.4), and there were frequent reports of altered serotonin (5-HT) and dopamine indices within the monoaminergic articles themselves, in addition to transporter/receptor measures (such as SERT, 5-HT\_2_A binding). Directions differed by matrix and method; for example, some datasets reported higher dopamine-linked measures, while SERT-related reports were more often reduced in OCD [44,45,46,47,48,49].

#### 3.1.4. Immune and Inflammatory Activation in OCD

Cytokines and acute-phase proteins were frequently evaluated. IL-6 was among the most common targets and was often reported as elevated. TNF-α and CRP were also frequently increased, though some reports noted heterogeneity in direction or non-significant findings depending on matrix and adjustment Where assessed, IL-1β and IL-17 tended to be higher in patients with OCD [50,51,52,53].

#### 3.1.5. Oxidative Stress and Mitochondrial Dysfunction in OCD

Regarding the oxidative pathways, markers of lipid peroxidation and antioxidant defenses were bold. MDA was repeatedly elevated across datasets that measured it. Glutathione (GSH or total) and superoxide dismutase (SOD) were commonly reduced where reported. A smaller number of studies assessed catalase and other redox enzymes, overall indicating diminished antioxidant capacity. Direct mitochondrial functional assessments were limited in this dataset; when present, findings aligned with heightened oxidative stress [54,55,56,57].

#### 3.1.6. HPA Axis Dysregulation and Neuroendocrine Changes in OCD

Cortisol was the most commonly assessed neuroendocrine marker. Across studies that measured it, cortisol was frequently higher in OCD, though directionality varied across matrices (plasma/serum, saliva, hair) and sampling paradigms. DHEA was less frequently reported, with some datasets mentioning increases [58,59,60].

#### 3.1.7. Epigenetic Modifications and Gene Expression Changes in OCD

This domain included DNA methylation and expression studies for neurotransmission and stress-related genes. The higher SLC6A4 (transporter of serotonin) methylation was commonly reported where measured; a higher level of methylation of the BDNF was also reported in some studies. Some studies also reported some additional loci such as OXTR, MAOA, and NR3C1, with direction of change varying by studies. Additionally, expression data such as OXTR mRNA complemented the methylation findings in some results [61,62,63].

#### 3.1.8. Neurotrophic Factor Alterations and Neuroplasticity Changes in OCD

BDNF protein was the most frequently studied neurotrophic factor. Where direction was specified, BDNF was often lower in OCD samples, although inverse or null findings were also present across smaller datasets. A smaller number of reports addressed IGF-1, NGF, and related plasticity-linked factors, with heterogeneous results [64,65,66,67,68].

#### 3.1.9. Peripheral Metabolic Alterations and Tryptophan-Kynurenine Pathway Changes in OCD

This category captured tryptophan-kynurenine pathway measures and other metabolic signals. Tryptophan was commonly reduced in datasets that quantified it, while kynurenine and quinolinic acid were frequently reported as elevated. Outside of the Trp-Kyn axis, leptin and other endocrine/metabolic markers were inconsistently altered, with direction varying by study [69,70,71,72].

### 3.2. Step 2: Gut Microbiome Alterations and Interventions in OCD

#### 3.2.1. Study Selection and Overview

Twenty studies met step 2 eligibility (Figure 2). Six were human (four stool microbiome cohorts, one serum barrier-protein study without microbiome, and one Mendelian randomization), and fourteen were animal models spanning causal manipulations such as FMT or antibiotics, probiotic interventions, diet, metabolic perturbations, genetic, and observational designs.

#### 3.2.2. Human Gut Microbiome Studies in OCD Patients

**Diversity**: Findings across the four stool cohorts were mixed. In a 16S dataset, Turna et al. reported lower Inverse Simpson in OCD, with no beta-diversity separation [53]. Domènech et al. observed a trend in alpha diversity that did not survive Bonferroni correction and no beta-diversity difference [11]. In shotgun metagenomics, Zhang et al. reported lower Shannon (*p* = 0.029) and lower Chao1 (*p* = 0.022) in OCD, while Simpson did not differ; groups exhibited only minor separation in species composition according to beta diversity indices (Bray–Curtis PERMANOVA R^2^ ≈ 0.03, *p* = 0.001, weighted UniFrac R^2^ ≈ 0.03, *p* = 0.001) [73]. In contrast, Chen et al. found no differences at baseline or post-therapy for richness, Shannon, Faith’s PD, or Bray–Curtis/UniFrac metrics [74]. Notably, clinical symptoms improved following ERP in this cohort, without significant microbiome change, highlighting a dissociation between clinical response and stool community metrics in the available data.

**Differential taxa and functions:** Evaluating the human-only, adjusted, FDR-significant, same method class studies, no taxon replicated across cohorts. Despite originating from single studies, the signals consistently pointed toward butyrate producers. In Zhang et al. (shotgun), OCD showed FDR-significant decreases in species such as *Faecalibacterium prausnitzii*, *Eubacterium rectale*, *Ruminococcus bromii*, *Roseburia faecis*, and *Bacteroides plebeius*, with increases in *Bacteroides stercoris*, *B. uniformis*, and *Parabacteroides distasonis* [19]. Functional profiling indicated reduced butyrate biosynthesis and tryptophan metabolism, alongside higher LPS biosynthesis (FDR-significant). Turna et al. reported lower *Oscillospira*, *Odoribacter*, and *Anaerostipes* in OCD [53]. Domènech et al. provided a LEfSe list (exploratory) and noted a single negative correlation between *Lachnospira pectinoschiza* and Y-BOCS [11].

**Biomarkers and genetics linked to a gut axis:** Two human papers bridged pathways related to the microbiome without stool sequencing. Zengil et al. reported elevated serum zonulin and occludin in OCD, each positively correlated with symptom severity (zonulin r = 0.514; occludin r = 0.623) [75]. He et al. used Mendelian randomization (MiBioGen taxa as exposures; OCD as outcome) and reported several nominally significant associations between gut taxa and OCD such as lower Proteobacteria associated with lower OCD odds [76]. Finally, Zhang et al. quantified stool metabolites and observed lower acetate, propionate, and butyrate in OCD [73]; in Chen et al., dietary fiber intake was lower in OCD (q = 0.04) despite null microbiome differences [74] (Table 1).

**Risk of bias assessment:** We appraised the seven studies of the second step that had human participants, using the JBI checklists [37]. Overall, two studies were low risk, one moderate, one moderate to high, and two high for their human components. Turna 2020 and Chen 2023 were judged low risk since they both used clear diagnostic ascertainment and matched controls, prespecified exclusions, described laboratory and bioinformatic methods, and employed appropriate statistics with multiplicity control [53,74]; Chen also recorded diet and handled follow-up within a longitudinal design [74]. Domènech 2022 was moderate risk since their case–control definitions and sequencing methods were adequate with multiple testing controls, but confounder identification and adjustment beyond basic matching such as diet, BMI, and medications were insufficiently explained [11]. Zengil 2024 was assessed moderate to high risk because selection and matching procedures and confounder control were not clearly reported despite valid assay methods and appropriate basic statistics [75]. The human donor comparison in Zhang 2024 was rated high risk; although donors were first episode and drug naïve, the very small, selected donor set (four OCD and four control) and limited detail on control screening and covariate handling restricted internal validity [73]. Primary inferences in that paper derive from the animal FMT arm (outside the JBI scope). He 2025 [76] was assessed as high overall risk according to an MR-specific checklist; while instrumental strength, independence from confounders, assessments of pleiotropy, heterogeneity, and population structure were all rated as low risk. The analysis screened approximately 200 taxa using nominal significance thresholds with limited correction for multiple testing. As a result there is an increased chance of a spurious findings despite supportive sensitivity analysis [76]. In total, selection, comparability procedures and confounder control were the main vulnerabilities across studies (Table 2).

#### 3.2.3. Preclinical Models: Microbiome Manipulation in OCD-like Behaviors

**Causal manipulations (FMT/antibiotics)*:*** Converging evidence indicates that gut microbes can drive repetitive or compulsive-like behavior in rodents. In Zhang et al. study, germ-free mice receiving FMT from an OCD participant showed more grooming (*p* < 0.01) and greater locomotion (*p* < 0.05) than mice receiving FMT from healthy donors; short-chain fatty acid supplementation partially rescued this behavior [73]. In an immunogenetic model, Cox et al. used antibiotics and FMT to establish microbiome causality, and further showed that microbial metabolites such as hippurate (HIP) and 3-PP could induce repetitive behavior via the D1-receptor mechanism [77]. A study regarding maternal microbiome and diet reported higher marble burying in male offspring in addition to 16S-based OTU shifts [78]. In addition, a brief abstract noted that antibiotics rescued repetitive behavior in zonulin-transgenic mice [79].

**Probiotic interventions:** Three models reported behavioral attenuation. In Ghuge et al. (quinpirole model, 16S profiled), an 8-week multistrain probiotic prevented increased marble burying and self-grooming and improved elevated plus maze (EPM) performance, amygdala IL-6 and CRP mRNA elevations with quinpirole were prevented, frontal-cortex nNOS was reduced, colon goblet cells and villus/crypt ratio improved, and alpha/beta diversity did not differ across groups [80]. Sanikhani et al. found improved open-field exploratory behavior with Lactobacillus casei Shirota and higher OFC BDNF [81]; Kantak et al. showed attenuation of stereotypy, marble burying, and center-distance deficits with Lactobacillus rhamnosus GG pretreatment [82].

**Diet and metabolic perturbations:** In a colitis sensitized model, Zhang et al. 2020 reported that time-restricted and intermittent energy-restriction regimens improved anxiety-like behavior (EPM test) and reduced marble burying, with stool 16S shifts (PLS-DA/LEfSe), lower serum LPS, lower colonic TNF-α, IL-1β, MDA, improved brain oxidative markers, and higher stool SCFAs [19]. Fortunato et al. found an altered offspring microbiome, changes in tryptophan and 5-HT biomarkers, and sex-specific behavioral effects [83]. Merchán et al. reported animals with more compulsive licking generally had more prominent fecal DGGE diversity changes and lower plasma 5-HT. And Deng et al., 2022 (HFD ± metformin) reported stool 16S shifts with diet and improvements in marble burying and grooming alongside 5-HT pathway biomarker changes with metformin [84].

**Baseline/genetic or observational models:** Two models underscored null or subtle microbiome differences. Despite altered gut function, Wilson et al. reported no baseline stool alpha or beta differences vs. wild-type [26]. Scheepers et al. found beta-diversity separation between large nest building and normal nest building groups using CLR/Aitchison PERMANOVA (*p* < 0.05), with no alpha-diversity differences in deer mice [85]. Jung et al. identified OTU-level drug × injection interactions across repeated exposures [86]. D’Addario et al. observed lower total SCFAs and butyrate at an early timepoint, with increases in stereotyped behaviors [64]. Together with SAPAP3 and deer mouse (β separated), these models specify the baseline range against which effects of FMT, ABX, and probiotic are interpreted (Table 3).

**Risk of bias assessment**: We appraised the 14 studies with animal cases using SYRCLE’s risk of bias tool [38]. Across domains, methodological reporting was limited, resulting in many judgments of unclear risk. None of the studies explicitly described random sequence generation, allocation concealment, or random housing, which prevented ratings of low risk in these areas. Baseline comparability was mostly enough since most experiments used inbred strains of the same age and sex, although small sample sizes in some studies increased vulnerability to imbalance [83]. The most frequent high-risk domain was outcome assessment, as behavioral testing was typically scored by investigators without blinding. However, in animal behavior studies, blinding is often difficult or not feasible, especially when phenotypes are bold, so this domain warrants cautious interpretation rather than being assumed to be a methodological flaw. Attrition and selective reporting were generally handled, with most studies reporting full outcomes aligned with their methods. Additional risks included small maternal groups [83] and use of post hoc phenotyping [85]. In total, the animal studies are best characterized as having consistent strengths in complete reporting and baseline standardization, but frequent weaknesses in randomization, concealment, and feasibility of blinding for behavioral outcomes (Table 4).

### 3.3. Bridge to Step 1 Biomarker Axes

Several domains of step 1 reappear here with direct microbiome context. Barrier integrity is implicated by elevated serum zonulin/occludin in OCD (with positive severity correlations) and by antibiotic rescue in zonulin-transgenic mice [75,79]; probiotic and dietary studies report improvements in the colon histology [19,80]. The SCFA axis shows lower stool SCFAs in human OCD [73], and reductions in stressed rodents [64], while SCFA supplementation partly rescues behavior after OCD-donor FMT [73]. Signals along the endotoxin/inflammation axis include lower serum LPS under gut-targeted fasting regimens [19], elevated CRP in human OCD [53], and prevention of higher IL-6 and CRP mRNA with probiotics [80]. Tryptophan/monoamines appear in human shotgun functional shifts, indicating lower tryptophan metabolism [73], and in maternal diet or metabolic models affecting 5-HT biomarkers [83].

Notably, despite multiple signals in some studies, no human taxon replicated under the pre-specified criteria (adjusted, FDR-significant, same method). Key domains remain under-evaluated in human cohorts, in particular domains regarding glutamatergic and GABAergic, HPA axis, oxidative, nitrosative, mitochondrial, cytokine panels (other than CRP), LBP/sCD14, neurotrophins, and epigenetic.

### 3.4. Mechanistic Causal Triangulation Across Microbiome-OCD Axes

We organized the results of step 2 into a ladder of causality: (a) human observations, (b) transfer of phenotype by human to mouse FMT, (c) depletion by antibiotics or transfer via FMT in animal models, and (d) rescue by targeted metabolites, probiotics, or diet within each mechanistic axis. Results are reported exactly as in the included studies; when statistics were not provided, none are inferred.

#### 3.4.1. SCFAs/Microbial Metabolites

Human observations in drug-naïve OCD patients using shotgun metagenomics revealed lower Shannon and Chao1 diversity, FDR-significant reductions in butyrate-producing species, decreased butyrate biosynthesis and tryptophan-metabolism pathways, as well as reduced stool acetate, propionate, and butyrate [73].

Transferring fecal microbiota from OCD donors to germ-free mice increased grooming and locomotion compared with FMT from healthy donors. Notably, SCFA supplementation partially rescued these behaviors [73].

Additional findings show that social isolation decreased total fecal SCFAs and butyrate at early timepoints [64], while time-restricted or intermittent energy-restriction regimens increased stool SCFAs and improved behavior in a dextran sodium sulfate (DSS)-sensitized paradigm [19]. Across models, restoration of SCFA tone consistently shifted behavior toward control levels.

#### 3.4.2. Endotoxin/Barrier Integrity

Human observations: Clinical studies report elevated serum zonulin and occludin levels in individuals with OCD, with both biomarkers showing positive correlations with symptom severity. Shotgun metagenomic functional profiling further indicated enrichment of lipopolysaccharide (LPS) biosynthesis pathways [73,75].

Also, depletion and transfer reports showed that in zonulin-transgenic mice, antibiotic treatment ameliorated repetitive behaviors [19,79].

Regarding rescue and concomitant biomarker changes, in a DSS-sensitized model, time-restricted or intermittent feeding regimens reduced serum LPS concentrations, improved colonic histology, and attenuated anxiety-like and repetitive behaviors in the EPM and marble-burying tests [73,75]. Similarly, administration of a multistrain probiotic enhanced goblet cell abundance and villus-to-crypt ratio in the colon, accompanied by behavioral improvements [80].

In addition, across models, gut-targeted interventions that lower endotoxin exposure and restore intestinal barrier integrity are consistently associated with reductions in repetitive and anxiety-like behaviors [19,79].

#### 3.4.3. Monoaminergic Signaling (5-HT/DA)

Human observational studies have identified reduced tryptophan metabolism within functional pathways sequenced by shotgun in patients with OCD [73]. In immunogenetic models, microbiome causality was demonstrated through antibiotic or FMT interventions, while specific microbial metabolites such as HIP and 3-PP were shown to induce repetitive behaviors via a D1-receptor mechanism [77]. Under high fat diet conditions, metformin treatment improved marble-burying and grooming behaviors, alongside restoring markers of the 5-HT pathway [84]. Maternal dietary tryptophan intake altered offspring microbiome composition and tryptophan/5-HT related biomarkers, producing behavioral outcomes specific to sex [83]. In quinpirole-induced models, probiotic administration was associated with behavioral improvements and modulation of monoaminergic gene expression, including HTR2A and SLC6A4 [81]. Collectively, these findings indicate that manipulation of the microbiome linked to monoaminergic pathways can modify repetitive behavioral phenotypes across multiple experimental models (Table 5).

## 4. Discussion

In this two-step evidence synthesis, we aimed to integrate 357 biological studies of OCD with 20 microbiome-focused studies to unsnarl the intertwined intersects of gut–brain axis and OCD. Across heterogeneous human cohorts, community-level microbiome differences were small and inconsistent, whereas function-level results and host biomarkers intersected on three axes, (i) relative SCFA deficiency, (ii) barrier dysfunction with endotoxin-adjacent signaling, and (iii) monoaminergic pathway engagement. These human signals align with preclinical transfer and rescue studies and are further supported by genetic causal inference. Together, they suggest the gut microbiota acts as a modifiable co-pathway in a subset of patients rather than an independent etiology.

### 4.1. Integrating OCD Biology with Gut–Brain Mechanisms

In general, cross-sectional and longitudinal studies fail to show robust α/β diversity separations between OCD and controls after appropriate statistics and covariate control, although isolated α-diversity reductions are reported in smaller datasets. For example, a well-designed longitudinal shotgun study detected no baseline or post-treatment α/β diversity differences, while documenting lower dietary fiber intake in OCD stand-alone (66). In contrast, function-level signals recur across studies. Pathway analysis reveals reduced butyrate biosynthesis and tryptophan metabolism with enrichment of LPS biosynthesis in OCD, alongside lower stool SCFAs in case samples [73]. At the human level, case–control studies find higher circulating proteins related to tight junctions (zonulin and occludin) in OCD, positively correlated with symptom burden [75]; low-grade inflammation is also observed, including higher CRP that correlates with Y-BOCS severity [53]. Collectively, the evidence suggests that changes in microbiomes’ metabolic functions are more consistent across studies than changes in bacterial genus, and these functional shifts align with host markers of barrier disruption and inflammation.

Additionally, Mendelian randomization studies suggest that some broad groups of bacteria such as *Bacillales*, Eubacterium *ruminantium* group, and *Lachnospiraceae* UCG001 may contribute to OCD risk, while others such as *Ruminococcaceae* and *Bilophila* may be protective (69). Although current methods cannot identify all strains, these findings support a causal influence of the microbiota on OCD at higher taxonomic levels.

### 4.2. Convergent Pathways: From Microbiome Dysfunction to OCD Symptoms

#### 4.2.1. SCFA Deficiency and Intestinal Barrier Dysfunction in OCD Pathogenesis

Human studies show reduced capacity for butyrate production and lower stool SCFA levels in OCD, which occur alongside modest increases in inflammation [73]. In germ-free mice, FMT from drug-naïve OCD patients induces repetitive and anxiety-like behaviors, while SCFA supplementation partly reverses these effects. Together, these findings suggest that SCFAs are not just correlational markers but may play a direct role in OCD-related mechanisms [73].

#### 4.2.2. Endotoxin-Mediated Barrier Disruption and Neuroinflammation in OCD

Clinically, serum zonulin and occludin are elevated in OCD and correlate with symptom severity [19]. In parallel, metagenomic analyses show increased pathways related to LPS biosynthesis in OCD. In animal models, inducing gut permeability or endotoxemia leads to compulsive-like behaviors, while interventions that strengthen the gut barrier or lower endotoxin levels, such as specific dietary approaches, reduce serum LPS, improve gut tissue health, and lessen anxiety-like and repetitive behaviors [73].

#### 4.2.3. Microbiome-Mediated Monoaminergic Dysfunction in OCD: Tryptophan-Kynurenine and Dopamine-Serotonin Pathways

Shotgun-based pathway analyses show alterations in tryptophan metabolism in OCD [73], with additional evidence from human and animal studies pointing to the oxytocin-receptor and the involvement of monoaminergics [64]. Experimental work demonstrates that microbial aromatic metabolites can influence dopamine-related behaviors; in one model, metabolites such as HIP and 3-phenylpropionate were sufficient to trigger repetitive behaviors through a D1-receptor mechanism.

### 4.3. Cross-Species Bridge: Metabolic Signatures

Metabolomic findings help link human data to animal models. In drug-naïve patients, serum succinic acid is elevated and correlates with symptoms. The same increase appears in serum and the medial prefrontal cortex of mice after colonization with OCD microbiota, strengthening the evidence for a causal connection between gut metabolism and brain circuits [73,74]

Sources of heterogeneity: Differences between studies can largely be justified by variations in patient groups, comorbidities, and study methods. Patients differ in their symptom type (such as contamination, cleaning, checking, harm), disease severity, and course. Common comorbidities such as depression, anxiety, and IBS-like symptoms also affect inflammation and measures related to the gut, adding further variability [64]. Medications such as SSRIs, antipsychotics, PPIs, and metformin can alter the microbiome [74]. Diet is another key factor; OCD patients often tend to lower fiber intake than controls. Yet most studies do not collect detailed dietary data [65,68]. Sex and hormonal influences remain understudied. Differences in sample sources also add variability; for example, oral studies report reduced Coprococcus (linked to DOPAC synthesis), which may not fully reflect gut changes [11]. Finally, many studies rely on small, single-site cohorts, limiting statistical power.

### 4.4. What Is Probably True Now

There are three aspects we can propose as mechanisms with high confidence. First, barrier and inflammation are involved in OCD. Circulating zonulin/occludin and low-grade inflammatory markers such as CRP are consistently elevated and scale with symptom severity [50,76,88].

Second, functional changes are more reliable than taxonomic shifts. Reduced capacity for SCFA production and increased LPS-related pathways are reproducible across studies, whereas genus- or family-level differences are inconsistent [73,81]. While reduced SCFA biosynthetic capacity appears to be among the most consistent functional findings across human OCD studies, this pattern should be interpreted with important caveats. Only a limited number of studies (N = 4) have directly measured stool SCFAs or SCFA-producing pathways in OCD patients, and sample sizes remain small (largest N = 64 total participants). Importantly, none of these studies implemented rigorous dietary control protocols, despite substantial evidence that fiber intake directly influences SCFA production. Given that OCD patients show significantly lower dietary fiber consumption than controls (mean 24.2 g vs. 34.9 g daily), the observed SCFA reductions may reflect secondary dietary effects rather than primary microbiome dysfunction. Furthermore, SCFA measurements are technically challenging due to rapid colonic absorption and require careful pre-analytical handling. The apparent consistency of SCFA-related findings may also reflect methodological homogeneity rather than biological replication, as studies have used similar populations (primarily treatment-seeking, medicated patients) and analytical approaches. Therefore, while SCFA deficiency represents a promising mechanistic pathway supported by animal transfer studies, definitive conclusions require larger, diet-controlled studies with standardized SCFA measurement protocols [89,90].

And third, microbiota can influence OCD-like behavior in animals. FMT from OCD donors induces repetitive and anxiety-like behaviors in mice, while SCFA supplementation and barrier-targeted strategies partially rescue these phenotypes [19,73,85].

With moderate confidence we can conclude that “high barrier/inflammation” subgroup may exist. Patients with elevated zonulin, occludin, or CRP appear more likely to benefit from interventions that are barrier targeted or SCFA restoring [75].

In addition, causality likely runs from microbiota to OCD at broad taxonomic levels. Mendelian randomization supports that microbiota correlates with OCD directionality for certain clades, although strain-specific targets remain undefined [76].

Finally, composite host–microbe panels may be useful. Multi-marker panels (zonulin or occludin + CRP ± stool SCFAs or microbial functions) appear more promising for stratification and monitoring than any single taxon [77,83].

And finally, two things can be assumed as emerging in the field. First, monoaminergic involvement may apply to a subset. Alterations in microbial metabolites linked to the tryptophan-kynurenine and dopamine-serotonin pathways likely contribute to symptoms in some patients [73]. And non-gut sites could add information but are not substitutes. Oropharyngeal and oral microbiome changes may provide complementary perspectives but require validation against gut and metabolite measures [11] (Figure 3a,b).

### 4.5. Critical Confounding Factors in Microbiome-OCD Research

The interpretation of microbiome findings in OCD is significantly complicated by multiple confounding factors that have received insufficient attention in existing studies, despite their established effects on gut microbial composition.

**Medication Effects**: Psychotropic medications commonly prescribed in OCD exert substantial effects on gut microbiota composition and function. SSRIs, the first-line treatment for OCD, significantly alter microbial diversity and specific taxa abundance, with studies showing both α-diversity reductions and specific enrichment of Bacteroides and depletion of Lachnospiraceae. Antipsychotics, frequently used for treatment-resistant OCD, demonstrate even more pronounced effects, consistently increasing potentially pathogenic taxa (Enterobacter, Klebsiella) while reducing beneficial SCFA-producers (Faecalibacterium, Akkermansia). Critically, these medication-induced changes occur within 4–8 weeks of treatment initiation, potentially masking or mimicking disease-related alterations. Most human OCD microbiome studies include predominantly medicated participants (typically 70–90%), without adequate control for medication effects, drug duration, or dose-dependent relationships [91,92].

**Dietary Patterns and Nutritional Status**: OCD patients exhibit distinct dietary patterns that directly influence microbiome composition independently of psychiatric symptoms. Reduced fiber intake is consistently documented in OCD patients, with deficits of approximately 30% compared to controls. This reduction is particularly problematic because fiber intake directly determines SCFA production capacity, with individual responses varying significantly based on baseline microbiota composition. Restrictive eating behaviors, common in OCD due to contamination fears or ritualistic patterns, can reduce overall dietary diversity and specific nutrient availability. Food avoidance related to contamination obsessions may systematically bias against certain food groups (fresh produce, fermented foods) that support beneficial microbiota. Importantly, dietary effects on microbiota occur rapidly (within 1–3 days) and can persist for weeks, making them difficult to control retrospectively [91,93].

**Comorbidity Interactions**: OCD frequently co-occurs with conditions that independently affect the gut microbiome. Depression (present in ~60% of OCD patients) is associated with reduced microbiota diversity and altered SCFA production, with effect sizes comparable to those reported in OCD. Anxiety disorders show overlapping microbiome alterations, including reduced Faecalibacterium and increased inflammatory taxa. Gastrointestinal symptoms (present in ~40% of OCD patients) may reflect primary gut dysfunction rather than secondary microbiome changes. The bidirectional relationship between psychiatric comorbidities and microbiome alterations makes it particularly challenging to isolate OCD-specific effects without large, well-phenotyped cohorts [74,94].

**Behavioral Factors Specific to OCD**: OCD symptoms themselves create behavioral patterns that influence microbiome composition through non-biological pathways. Excessive cleaning and sanitization behaviors may reduce environmental microbial exposure and alter skin–gut microbial transfer. Compulsive handwashing with antimicrobial products could affect oral microbiota and subsequently influence gut colonization. Avoidance behaviors may limit exposure to diverse environments and associated microbial communities. Sleep disruption, common in OCD, independently affects circadian regulation of gut microbiota. These behavioral factors create a complex feedback loop where OCD symptoms may drive microbiome changes that subsequently influence symptom severity [95,96].

**Methodological and Analytical Confounders**: Sample collection and storage protocols vary significantly across OCD microbiome studies, with different stabilization methods and storage durations that can alter apparent microbial composition. Analytical pipelines using different databases, clustering methods, and statistical approaches can yield discordant taxonomic results from identical samples. Population stratification by age, sex, geography, and socioeconomic status affects baseline microbiome composition but is inconsistently controlled across studies. Seasonal effects and circadian sampling timing introduce additional variability that may exceed disease-related effects in small studies [97,98].

**Clinical Implications**: These confounding factors have several critical implications for OCD microbiome research and clinical translation. First, medication-naïve studies are essential but challenging given that most OCD patients receive treatment before research participation. Second, dietary assessment and control should be mandatory in microbiome studies, ideally using standardized food frequency questionnaires or controlled feeding protocols. Third, longitudinal designs with pre-treatment baselines are necessary to distinguish disease-related from treatment-induced changes. Fourth, composite biomarker approaches incorporating host markers (zonulin, inflammatory markers) alongside microbiome measures may be more robust to confounding than microbiome-only analyses. Finally, stratified analyses by medication status, comorbidity profiles, and symptom subtypes are necessary to identify clinically actionable subgroups rather than population averages [99,100].

The extensive confounding in current OCD microbiome studies suggests that observed associations may reflect a complex mixture of disease-related, treatment-related, and behavioral factors rather than primary pathogenic mechanisms. Future studies must address these methodological challenges to establish clinically meaningful microbiome–OCD relationships.

## 5. Conclusions

This two-step synthesis links established OCD biology with gut–brain mechanisms. The most reproducible signals are functional, reduced SCFA capacity and increased LPS/barrier disruption, together with host markers of permeability and inflammation (higher zonulin/occludin and CRP). Taxonomic shifts are variable, but convergent animal and genetic evidence support a mechanistic contribution of the microbiota in at least a subset of patients. Near-term translation should emphasize biomarker-guided, mechanism-matched adjuncts to ERP/SSRI and evaluate success by coupling symptom change to shifts in SCFAs, barrier markers, and monoaminergic metabolites.

## Figures and Tables

**Figure 1 life-15-01585-f001:**
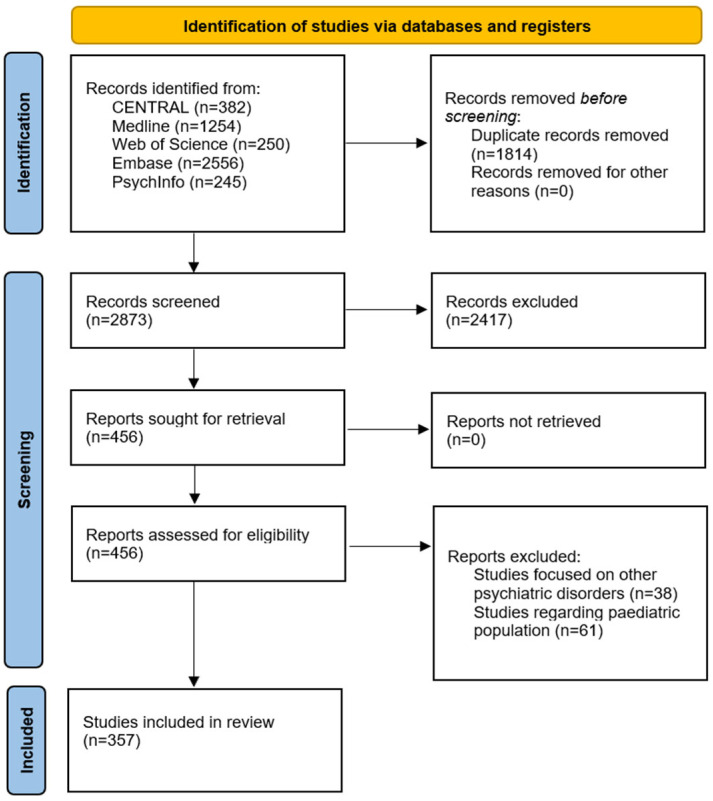
PRISMA flow diagram of step 1.

**Figure 2 life-15-01585-f002:**
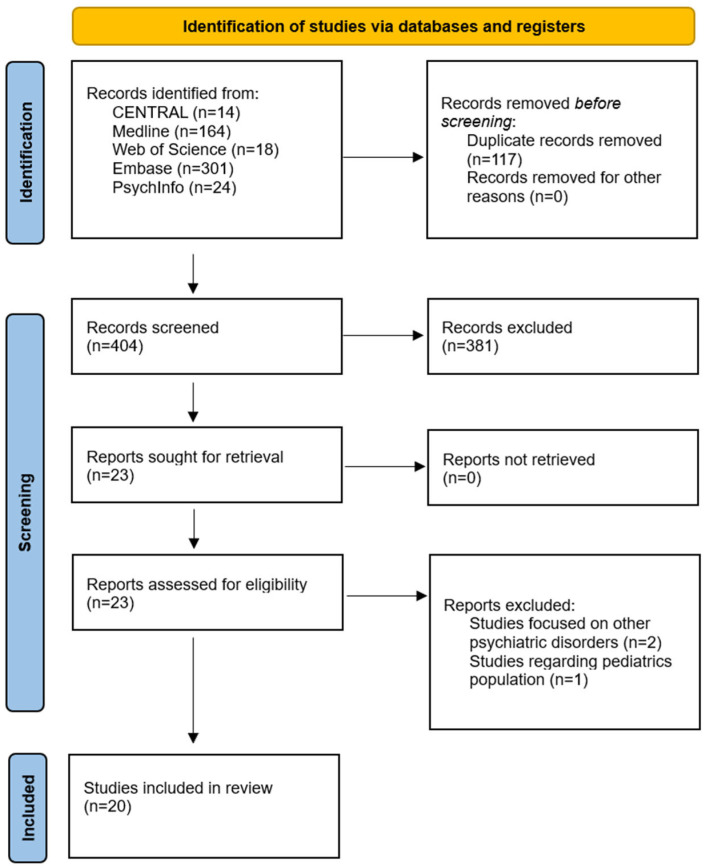
PRISMA flow diagram of step 2.

**Figure 3 life-15-01585-f003:**
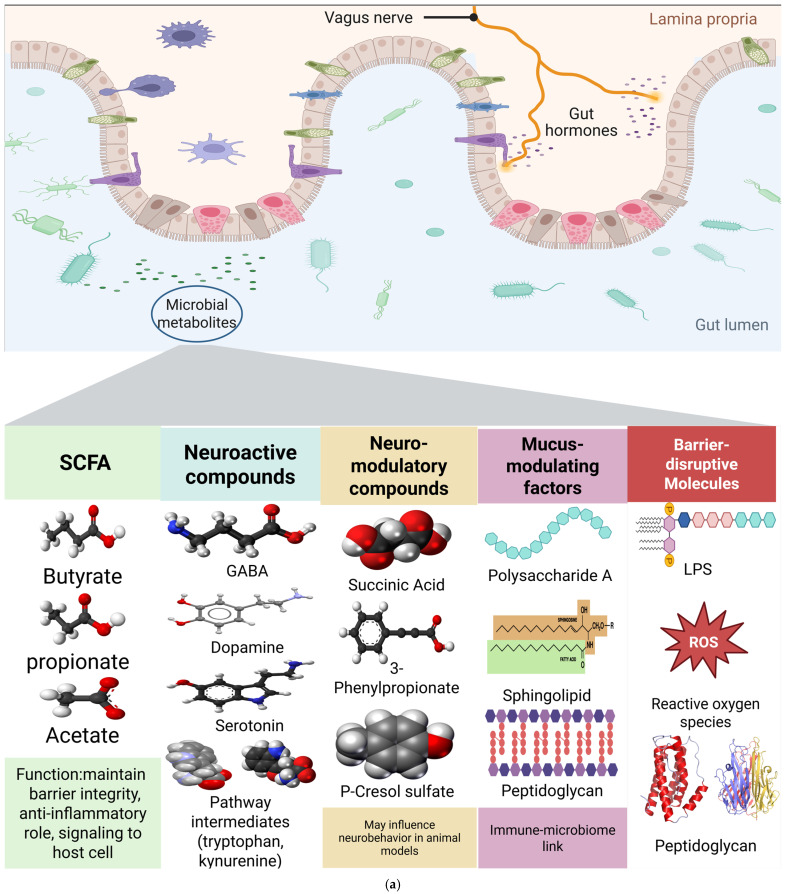
(**a**) Microbial metabolites produced by gut bacteria exert diverse effects on intestinal barrier integrity and host signaling pathways. The gut microbiome generates beneficial metabolites including SCFAs (butyrate, propionate, acetate) that maintain barrier integrity and provide anti-inflammatory signaling, neuroactive compounds (GABA, dopamine, serotonin) that influence vagal nerve signaling, and mucus-modulating factors (polysaccharide A, sphingolipid) that support epithelial function. Conversely, dysbiotic conditions promote production of barrier-disruptive molecules including LPS, reactive oxygen species, and peptidoglycan fragments that compromise intestinal permeability. Molecular structures show the chemical diversity of these metabolites, with functional annotations indicating their roles in maintaining barrier integrity (green), providing neuroactive signaling (blue/yellow), supporting immune–microbiome interactions (pink), or disrupting barrier function (red). These metabolites serve as the biochemical foundation for gut–brain axis communication described in Figure 3b (↑ indicating increase, and ↓ indicating decrease; created in https://BioRender.com). (**b**) Systemic Gut–Brain Axis Mechanisms Leading to OCD Pathophysiology. Compromised intestinal barrier integrity enables translocation of microbial metabolites into systemic circulation, ultimately affecting brain regions implicated in OCD through neuroinflammatory and neurotransmitter pathways. Following production in the gut microenvironment (Figure 3a), metabolites cross the intestinal epithelium through three primary mechanisms: SCFAs traverse intact barriers to support systemic anti-inflammatory responses, LPS breaches compromised barriers (characterized by reduced zonulin and occludin expression) leading to systemic inflammation, and GABA/neuroactive compounds signal via vagal nerve pathways using bidirectional communication. These circulating factors reach key brain regions where they induce region-specific molecular alterations: the prefrontal cortex shows elevated pro-inflammatory markers (↑IL-6, ↑TNF-α) and microglial activation; the striatum exhibits neurotransmitter dysfunction (↓GABA/↑glutamate, ↓serotonin); and the thalamus demonstrates oxidative stress markers (↑MDA, ↓GSH). Convergent dysfunction across these CSTC circuit components results in the characteristic behavioral phenotypes of OCD such as circuit dysfunction, elevated stress biomarkers, repetitive behaviors, and impulsivity (↑ indicating increase, and ↓ indicating decrease; created in https://BioRender.com).

**Table 1 life-15-01585-t001:** Table of included human studies.

Study	Design/Cohort	n (OCD/HC)	Specimen	Method (16S/Shotgun/Other)	α Diversity (per Index)	β Diversity (Metric; PERMANOVA)	FDR-Sig Taxa (Direction)	Gut-Axis Biomarkers (Assay)	Clinical Link	Notes
He 2025 [76]	Two-sample Mendelian randomization (exposure: gut taxa GWAS; outcome: OCD GWAS)	MiBioGen exposure: 18,340 across 211 taxa; OCD outcome: FinnGen Europeans N = 199,169	Summary GWAS	MR	—	—	Positive causal signals reported for Bacillales, *Eubacterium ruminantium* group, Lachnospiraceae UCG001; protective for Ruminococcaceae and Bilophila (IVW-anchored, sensitivity tested)	—	—	Data sources and significance handling described; authors note potential weakness using *p* < 1 × 10^−5^ instruments in microbiome GWAS
Zhang 2024 [73]	Human → mouse FMT mechanistic study; donors drug-naïve	FMT donors: 4/4; separate serum cohort: 32/32	Mouse stool (post-FMT); human serum	16S full-length (mouse); targeted metabolomics (human serum)	—	Mouse FMT groups show compositional separation (methods/results narrative)	—	Succinic acid (SA) increased in OCD patients’ serum (*p =* 0.0016; N = 32/32); SA increased in OCD-colonized mice serum and mPFC; SA correlates with OCI-R in humans (R^2^ = 0.1278; *p =* 0.0037)	SA-OCI-R positive correlation; SA associated with higher Gammaproteobacteria and Clostridia, lower Bacilli (joint analysis)	Methods list full-length 16S for mice; human serum targeted LC-MS described; donor feces used for FMT
Zengil 2024 [75]	Case–control; adults	60/30	Serum	Other (barrier proteins)	—	—	—	Zonulin and occludin markedly increased in OCD; both correlate with Y-BOCS and duration (positive)	Positive correlations (Y-BOCS/HDRS)	No microbiota profiling performed.
Chen 2023 [74]	Longitudinal case–control with ERP follow-up	32/32	Stool	Shotgun WGS	Richness/Shannon/Faith’s PD: no differences at baseline; also, ns post-ERP (exact W and p in text)	Bray–Curtis, unweighted and weighted UniFrac: ns (F ≈ 0.02; *p* ≥ 0.50–0.97)	None	Fiber intake lower in OCD (mean 24.22 g vs. 34.90 g; q = 0.04)	Marked symptom improvement after ERP (Y-BOCS and OCI-R reductions reported)	Sequencing noted as WGS in Methods; diet assessed via FFQ linked to Swedish DB
D’Addario 2022 [64]	Case–control (adults)	64/51	PBMCs; saliva	Other (OXTR mRNA and DNA methylation in PBMCs; OXTR methylation in saliva; saliva phyla by rRNA gene PCR)	—	—	(Saliva phyla abundances quantified; not a 16S survey)	PBMC OXTR mRNA/methylation measured; saliva OXTR methylation; saliva phyla (Actinobacteria, Firmicutes, Fusobacteria, Bacteroidetes, Proteobacteria)	No correlation with Y-BOCS found in saliva methylation analysis; meds stable ≥1 month in many OCD participants; some confounders not recorded	—
Domènech 2022 [11]	Case–control + within-OCD pre/post (paired for a subset)	38/33 (controls)	Stool and oropharyngeal	16S rRNA amplicon	Indices computed: Shannon, Chao1, Observed, Faith’s PD, Evenness-methods	PERMANOVA with multiple distance metrics (999 perms-methods)	—	—	—	Cohort make-up: baseline 54 stool and 62 oropharyngeal samples; 28 OCD had paired T0/T3; +7 single-timepoint OCD; 33 HCs
Turna 2020 [53]	Cross-sectional case–control (adults)	21/22	Stool	16S rRNA amplicon	Inverse Simpson decreased in OCD	No between-group separation in Jaccard/Bray–Curtis/weighted and unweighted UniFrac	Lower *Oscillospira* (OTU13), *Odoribacter* (OTU92), *Anaerostipes* (OTU137); W-stats shown; indicates FDR *p* < 0.02	CRP increased in OCD vs. HC.	CRP correlated with Y-BOCS.	—

**Table 2 life-15-01585-t002:** Risk of bias assessment of the included human studies using JBI’s assessment tool.

Human Case–Control Studies																		
Study (Year)	I1. Case Definition Adequate	I2. Control Definition Appropriate		I3. Case Selection Appropriate		I4. Control Selection Appropriate		I5. Matching/Comparability		I6. Exposure Measured Validly		I7. Same Exposure Measurement for Groups		I8. Confounders Identified	I9. Strategies for Confounding		I10. Appropriate Statistics	Overall RoB
Zengil 2024 [75]	Unclear	Unclear		Unclear		Unclear		Unclear		Yes		Yes		Unclear	Unclear		Yes	Moderate–High
Zhang 2024 [73]	Yes	Unclear		Yes		Unclear		Yes		Yes		Yes		Unclear	Unclear		Unclear	High
Chen 2023 [74]	Yes	Yes		Yes		Yes		Yes		Yes		Yes		Yes	Yes		Yes	Low
Domènech 2022 [11]	Yes	Yes		Yes		Yes		Yes		Yes		Yes		Yes	Unclear		Yes	Moderate
D’Addario 2022 [64]	Yes	Yes		Yes		Yes		Yes		Yes		Yes		Partially	No		Yes	Moderate
Turna 2020 [53]	Yes	Yes		Yes		Yes		Yes		Yes		Yes		Yes	Yes		Yes	Low
**Human Mendelian randomization** **study**																		
**Study**	**Assessment Tool**		**Instrument Relevance**		**Independence from Confounders**		**Exclusion Restriction/Pleiotropy**		**Heterogeneity (Cochran’s Q)**		**Population Stratification/Sample Overlap**		**Multiple Testing Across Taxa**			**Sensitivity Analyses**		**Overall Risk of Bias**
He 2025 [76]	MR-specific		Low		Low		Low		Low		Low		High			Low		High

**Table 3 life-15-01585-t003:** Table of included animal studies (↑ indicating increase, and ↓ indicating decrease).

Study	Model/Strain/Sex	Gut Manipulation	Microbiome Profiling	Behavior (Direction)	Gut-Axis Biomarkers	α/β Diversity (If Reported)	Notes
Fortunato 2025 [83]	Maternal tryptophan (Trp)-enriched diet (1.5% vs. 0.7% control) altered offspring gut microbiome and produced sex-specific behaviors: ↑ repetitive behavior (marble burying) in males; ↑ anxiety-like behavior in females; associative gut–brain axis evidence (no FMT/ABX).	None	16S rRNA amplicon	—	LEfSe families (sex-specific): Female TRP+ ↑ Tannerellaceae (0.049 ± 0.008 vs. 0.010 ± 0.003; *p* < 0.0001), ↑ Muribaculaceae (0.154 ± 0.030 vs. 0.092 ± 0.031; *p* = 0.0243), ↑ Eubacterium (0.009 ± 0.004 vs. 1.85 × 10^−4^ ± 1.61 × 10^−4^; *p* = 0.0075); ↓ Marinifilaceae (4.93 × 10^−4^ ± 0.001 vs. 0.024 ± 0.006; *p* = 0.0071), ↓ Saccharimonadaceae (0.002 ± 4.52 × 10^−4^ vs. 0.041 ± 0.028; *p* = 0.039), ↓ Oscillospiraceae (0.025 ± 0.007 vs. 0.063 ± 0.002; *p* = 0.0014). TRP+ sex-dimorphism: Bacteroidaceae higher in males than females (0.144 ± 0.050 vs. 0.062 ± 0.026; *p* = 0.0409); Monoglobaceae higher in females (*p* = 0.0170).	β: Females: Shannon CTR 4.84 ± 0.12 vs. TRP+ 4.31 ± 0.06 (*p* = 0.0358); richness (Chao) ns. Sex effect on Shannon in CTR lost in TRP+.	—
Cox 2024 [77]	γδ T-cell–deficient (TCRδ−/−) mice display microbiota-dependent repetitive/compulsive (marble-burying) behavior; microbiota transfer induces/abolishes phenotype; direct gut microbiome ↔ compulsive-like link.	FMT; Antibiotic	16S rRNA amplicon	—	Cecum: 3-phenylpropionate (3-PP) ↑ (~50×) in γδ−/−; 3-(4-hydroxyphenyl) propionate ↑ (~30×) in WT; Serum/CSF: HIP and p-cresol sulfate (pCS) ↑ in γδ−/−; HIP ↑ ~8× in CSF; pCS ↑ ~4× (Welch’s t-tests, *p* < 0.05).	α: Alpha: no differences between groups after colonization. | β: Beta: unweighted UniFrac; ADONIS significant by microbiota treatment, not genotype.	—
Wilson 2024 [26]	SAPAP3 knockout (validated compulsive/OCD-like model); gut function assessed plus baseline stool microbiome (16S) in SH mice; WT vs. KO showed no dysbiosis (species richness, alpha and beta diversity equivalent).	None (environmental housing; not microbiome-targeted)	16S rRNA amplicon (baseline SH subset)		Fecal water content ↑ in KO vs. WT (estimate 9.67, SE 3.13, t = 3.09, *p* = 0.005); fecal output ↑ in KO vs. WT (*p* = 0.045); gut permeability (FITC-dextran) trend ↑ in KO (*p* = 0.065); gut transit time slower in EE vs. SH (coef −0.66, exp(coef) = 0.52, *p* = 0.033).	Alpha diversity: no KO vs. WT difference. Beta diversity: no KO vs. WT difference.	—
Zhang 2024 [73]	Human stool from drug-naïve OCD vs. healthy; FMT into germ-free mice altered compulsive-like grooming/locomotion	FMT	Shotgun	Mouse FMT (OCD donor): ↑ grooming time (*p* < 0.01), ↑ locomotor activity (*p* < 0.05) vs. HC FMT; partial rescue with SCFA supplementation	Pathways (HUMAnN3, FDR < 0.05): ↓ butyrate biosynthesis, ↓ tryptophan metabolism, ↑ LPS biosynthesis; SCFA (stool acetate, propionate, butyrate) ↓ in OCD (*p* < 0.05, GC–MS)	α: Shannon: ↓ in OCD vs. HC (*p* = 0.029); Simpson: ns; Chao1: ↓ (*p* = 0.022) | β: Bray–Curtis: PERMANOVA R^2^ = 0.034, *p* = 0.001; Weighted UniFrac: PERMANOVA R^2^ = 0.027, *p* = 0.001	Metagenomics + FMT causality; extensive taxa, pathway, SCFA and behavior data
D’Addario 2022 [64]	Human: DSM-5 OCD vs. HC with OXTR epigenetics and oral (saliva) phyla qPCR (not gut); Animal: social isolation rats with fecal phyla qPCR + SCFAs and stereotyped behaviors	None	Other	Animal behavior (ISO vs. CTRL): Open field center time ↓ (t = 4.31, *p* < 0.001); wall rearing ↑ (t = 9.56, *p* < 0.001); hole-board head dippings ↑ (t = 4.69, *p* < 0.001). Rat PFC Oxtr mRNA ↓ (0.55 ± 0.10 vs. 1.10 ± 0.20; *p* = 0.007)	SCFAs (rat feces, LC–MS): Total SCFAs ↓ at T1 (CTRL 38.92 ± 7.42 vs. ISO 20.83 ± 3.59; *p* = 0.049); Butyrate ↓ at T1 (11.65 ± 1.54 vs. 6.69 ± 1.35; *p* = 0.049). Human: OXTR mRNA (PBMC) ↓ (0.35 ± 0.05 vs. 1.10 ± 0.10; *p* < 0.0001); OXTR exon 3 methylation ↑ (PBMC avg CpGs 5.21 ± 0.32 vs. 3.74 ± 0.15; *p* = 0.006; saliva avg 4.43 ± 0.31 vs. 2.95 ± 0.27; *p* = 0.021)	—	Human microbiome = oral (excluded for human gut analyses). Animal part qualifies (associative; no ABX/FMT). Microbiome measured by phylum-qPCR, not 16S/shotgun; no α/β stats. Correlations: PFC Oxtr vs. behavior (e.g., center time r = 0.554, *p* = 0.035)
Deng 2022 [84]	HFD induces ↑ repetitive behavior (marble burying, self-grooming) vs. ND; metformin reverses; gut 16S profiles and gut 5-HT pathway (Trp, 5-HT, 5-HIAA; TPH1/SERT) measured; associative gut–brain axis mechanism proposed (no FMT/ABX).	None	16S rRNA amplicon	—	HFD vs. ND: ↑ Lactococcus, Trichococcus, Romboutsia, Faecalibaculum; HFD + Met: ↑ Intestinimonas, Lactobacillus reuteri (LDA ≥ 4); HFD/HFD + Met: Melainabacteria ↓; HFD + Met vs. HFD: Tenericutes ↑; F/B ratio ↑ in HFD, metformin ↑ Bacteroidetes.	β: Richness (Observed_OTUs) ↓ in HFD vs. ND (significant); Simpson ns.	—
Merchán 2021 [87]	Schedule-induced polydipsia (SIP) model of compulsivity; High-Drinkers (HD) vs. Low-Drinkers (LD) ± chronic tryptophan-free diet; fecal microbiota profiled; compulsive licking increased only in HD under TRP-depletion.	None (dietary TRP manipulation)	16S (PCR-DGGE fingerprinting)	Plasma 5-HT ↓with TRP depletion (treatment effect F1,24 = 10.754, *p* < 0.01); BDNF: no group/treatment effects; 5-HT ↔ BDNF: positive correlation r = +0.514 (*p* < 0.01; N = 27).	Bray–Curtis cluster analysis: HD T- animals form a distinct cluster; LD T+/LD T- cluster together; HD T+ separate sub-cluster. PL functional organization: Fo ~66% (HD T-, 62% (HD T+ and LD T+), 56% (LD T-).	—	Behavior (compulsive licking): TRP depletion increased total licks (only in HD) (group × treatment × session F5,120 = 2.529, *p* < 0.05); HD T negative more than HD T positive from session three (*p* < 0.01).
Sanikhani 2020 [81]	Quinpirole-induced OCD-like behaviors in Wistar rats; probiotic L. casei Shirota (1 × 10^9^ CFU/g, daily ×4 wks) post-induction; open-field compulsive-checking metrics reported	Probiotic	Other (no microbiome profiling)	Open-field: exploratory behavior improved with L. casei (*p* = 0.018) and L. casei + fluoxetine (*p* = 0.004); group difference in key-zone time profiles *p* = 0.03; direction = decrease in OCD-like preference for corner “home base” zones.	—	—	No gut microbiome sequencing; probiotic is gut targeted. Quinpirole 0.5 mg/kg i.p. twice weekly ×5 wks; five groups × 6 rats.
Scheepers 2020 [85]	Natural compulsive-like large nest building (LNB) vs. normal nest building (NNB) in deer mice; gut microbiome profiled	None	16S	—	—	β: Aitchison (genus-level CLR): PERMANOVA *p* < 0.05; PC1 = 13.77%; PC2 = 10.91%	Duplicate with: Single timepoint fecal sampling during first hour of dark cycle; 86 genera detected; minimum read count ≥10,000; median read depth 34,686 reads/sample; n per group includes three males and eight females
Zhang 2020 [19]	DSS-induced colitis in male C57BL/6 mice; EPM and obsessive–compulsive-like (marble burying) behaviors measured	None	16S	EPM: % open arm entries ↑ vs. DSS (TRF, IER; *p* < 0.01); MBT: marbles buried ↓ vs. DSS (TRF, IER; *p* < 0.05)	LPS (serum) ↓ (TRF, IER; *p* < 0.01; ELISA Xinle Biotech); TNF-α (colon) ↓ (TRF, IER; *p* < 0.01; ELISA Xinle Biotech); IL-1β (colon) ↓ (TRF, IER; *p* < 0.01; ELISA Xinle Biotech); MDA (colon) ↓ (TRF *p* < 0.01, IER *p* < 0.05; Nanjing Jiancheng kit); TNF-α, IL-1β, IL-6 (brain) ↓ (TRF, IER; *p* < 0.01; RT-PCR); MDA, GSSG ↓, GSH ↑ (brain; *p* < 0.01; commercial kits); SCFAs (acetate, butyrate, isobutyrate) ↑ (TRF, IER; *p* < 0.05; GC Shimadzu)	—	Colitis model with OCD-like component; ADF arm worsened survival/colitis and excluded from behavioral/molecular follow-up; only TRF and IER analyzed for microbiome and behavior outcomes
Miranda-Ribera 2019 [79]	Zonulin transgenic mice (Ztm) with increased gut permeability; females show pronounced repetitive behavior (marble burying); males show increased anxiety-like behavior; gut microbiota depletion with antibiotics rescues behavioral phenotype; direct microbiota–behavior link (animal).	Antibiotic	Not stated (stool microbiota composition assessed)	Antibiotic depletion: decreases zonulin and inflammatory markers in brain, normalizes BBB tight junctions, and rescues behavioral phenotype (no statistics provided).	Small-intestinal permeability ↑; BBB tight-junction genes dysregulated; brain inflammatory markers ↑ (qualitative per abstract).	—	Ztm show dysbiosis; behavioral tests included marble burying (repetitive/compulsive-like) and elevated zero maze (anxiety-like). Outcomes and microbiome details are qualitative because this is a conference abstract.
Jung 2018 [86]	Quinpirole-induced compulsive checking rat model; serial fecal 16S at injections 1, 5, 9	None	16S	At injection 9: all four compulsive-checking criteria differed QNP vs. saline (t-tests *p* ≤ 0.003); “time to next checking bout” shorter in QNP (t = 2.028, *p* = 0.037, 1-tail). Locomotion: distance ↑, 2SDE ↓, path stereotypy ↑ (all *p* < 0.001 at injection 9).	—	—	Preprint; Raw reads rarefied to 43,345/sample; feces collected ~55 min post-injection; no α/β diversity stats or PERMANOVA reported.
Bruce-Keller 2017 [78]	Maternal gut microbiota manipulated (antibiotic depletion → FMT from HFD vs. CD donors); male offspring from HFD-microbiota dams show ↑ stereotypical/compulsive marble burying; direct gut microbiome ↔ compulsive-like link (animal).	FMT; Antibiotic	16S rRNA amplicon	—	Pre-pregnancy dams: PERMANOVA (weighted UniFrac) F = 5.2, *p* = 0.009; Offspring females: PERMANOVA F = 2.51, *p* = 0.003; Offspring males: PERMANOVA F = 2.05, *p* = 0.034.	—	Male offspring from HFD-microbiota dams: marble burying increased (*p* < 0.01); Open field inner-zone time/entries ↓ (*p* < 0.01–0.001); sucrose preference ↑ (ad lib); fear-conditioning freezing ↓ (tone test)—females largely unaffected.
Kantak 2014 [82]	RU 24969–induced OCD-like behaviors; probiotic pretreatment tested	Probiotic	—	L. rhamnosus GG (ATCC 53103) 1 × 10^9^ CFU/day oral gavage for 2 or 4 wks; male BALB/cJ mice; RU 24969 10 mg/kg i.p.; behaviors scored 60–90 min post-injection; no microbiome data	—	—	—

**Table 4 life-15-01585-t004:** Risk of bias assessment of the included animal studies using SYRCLE’s assessment tool.

Study	Sequence Randomization	Baseline Characteristics	Allocation Concealment	Random Housing	Blinding of Investigators	Blinding of Outcome Assessment	Incomplete Outcome Data	Selective Outcome Reporting	Other Sources of Bias	Overall Risk of Bias
Furtunato 2025 [83]	Unclear	Unclear	Low risk	Low risk	Low risk	High risk	Low risk	Low risk	High risk	High risk
Wilson 2024 [26]	Unclear	Low risk	Unclear	Unclear	Low risk	High risk	Low risk	Low risk	Low risk	High
Cox 2024 [77]	Low risk	Low risk	Unclear	Unclear	Unclear	Low risk	Low risk	Low risk	Low risk	Moderate risk
Ghuge 2023 [80]	Unclear	Low risk	Unclear	Unclear	Unclear	High risk	Low risk	Low risk	Low risk	High risk
Deng 2022 [84]	Unclear	Low risk	Unclear	Unclear	High risk	Low risk	Low risk	Low risk	Low risk	High risk
D’Addario 2022 [64]	Unclear	Low risk	Unclear	Low risk	Low risk	High risk	Low risk	Low risk	Low risk	High
Merchán 2021 [87]	Unclear	Low risk	Unclear	Low risk	Low risk	High risk	Low risk	Low risk	Unclear	High
Sanikhani 2020 [81]	Unclear	Low risk	Unclear	Unclear	Unclear	High risk	Low risk	Low risk	Low risk	High
Scheepers 2020 [85]	Unclear	Unclear	Unclear	Low risk	Low risk	High risk	Low risk	Low risk	High risk	High
Zhang 2020 [19]	Unclear	Low risk	Unclear	Low risk	Low risk	High risk	Low risk	Low risk	Low risk	High
Miranda-Ribera 2019 [79]	Unclear	Low risk	Unclear	Unclear	Low risk	Low risk	High risk	Low risk	Low risk	High
Jung 2018 [86]	Unclear	Low risk	Unclear	Unclear	Low risk	Low risk	Low risk	Low risk	Low risk	Moderate risk
Bruce-Keller 2017 [78]	Low risk	Unclear	Unclear	Low risk	Unclear	High risk	Unclear	Low risk	Low risk	High risk
Kantak 2014 [82]	Unclear	Low risk	Unclear	Low risk	Unclear	High risk	Low risk	Low risk	Low risk	High

**Table 5 life-15-01585-t005:** Summary of mechanistic causal triangulation across microbiome–OCD axes (↑ indicating increase, and ↓ indicating decrease).

Axis	Human Observation	Transfer (Human → Mouse FMT)	Depletion/Transfer (ABX/FMT in Models)	Rescue (Metabolite/Probiotic/Diet)	Behavioral Direction
SCFAs	↓ stool SCFAs; ↓ butyrate-producer species; ↓ butyrate pathways [73]	OCD-FMT ↑ grooming/locomotion [73]	—	SCFAs partially rescue; TRF/IER ↑ SCFAs [19,73]	Toward control with SCFA restoration
Endotoxin/Barrier	↑ serum zonulin/occludin (severity-linked); ↑ LPS-biosynthesis functions [19,73,75]	—	ABX rescue in zonulin-Tg [79]	TRF/IER ↓ serum LPS; improved histology; probiotic histology gains [19,81]	Reduced repetitive and anxiety-like behavior
Monoaminergic	↓ tryptophan-metabolism pathways [73]	—	ABX/FMT causality; metabolites (HIP, 3-PP) induce behavior via D1R [77]	Metformin improves 5-HT-pathway markers and behavior; maternal Trp diet shifts biomarkers and behavior; probiotic with gene-expression changes [81,83,84]	Behavior modified with pathway-targeted interventions

## Data Availability

All data used is presented within the text or Appendix A.

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
