# Peer review of "Gut Microbiota and Obsessive–Compulsive Disorder: A Systematic Review of Mechanistic Links, Evidence from Human and Preclinical Studies, and Therapeutic Prospects"

_life, 2025, doi:10.3390/life15101585_

Round 1

Reviewer 1 Report

Comments and Suggestions for Authors

The current review is a timely systematic review integrating OCD with gut microbiota. The two-step strategy is innovative, and the manuscript is satisfactorily structured. The focus on functional pathways (SCFAs, barrier integrity, monoaminergic signaling) rather than purely taxonomic differences is a notable strength and thus enhances the translational potential of the review. However, several areas could benefit from clarification or expansion.

  1. Both “obsessive-compulsive disorder” and “OCD” have been included as keywords. It would be preferable to retain only one of them.
  2. In line with the title, the Introduction could benefit from additional background, particularly focusing on mechanistic insights. A clearer distinction between what is established in animal models versus what remains uncertain in humans would also be valuable.
  3. Line 81: The terms “DSM and ICD criteria” could be briefly defined to help readers unfamiliar with these diagnostic systems.
  4. While the Inclusion criteria are described in detail, the Exclusion criteria section could be expanded. For instance, did you exclude in vitro studies, preclinical data, or data from patients with psychiatric comorbidities?
  5. The manuscript states that data were collected up to September 2025. Please clarify whether all articles published in September 2025 were included in the search.
  6. Section headings such as “Glutamatergic / GABAergic”, “Monoaminergic”, and “Immune & Inflammatory” would be clearer if explicitly framed in the context of OCD.
  7. Line 283: The statement “Table 2. This is a table. Tables should be placed in the main text near their first citation” should be revised, and each table should be provided with a clear and explanatory caption. Please ensure that captions for all tables are sufficiently descriptive.
  1. The discussion effectively emphasizes functional findings over taxonomic shifts. However, some claims (e.g., SCFA deficiency as a reproducible hallmark) should be qualified by noting the small number of human studies and the variability in dietary control.
  2. The issue of confounding factors such as diet, psychotropic medication, and comorbidities is mentioned, but could be discussed in greater depth, given their major impact on microbiome studies.

    10. Consider adding a schematic diagram that visually summarizes the proposed gut–OCD mechanisms (SCFA deficiency, barrier disruption, monoaminergic alterations). This would improve accessibility for readers unfamiliar with the field.

Author Response

First and foremost, I want to thank the reviewers and the prestigious journal of Life for their timely and precise process.

Here is our response to the reviewer’s comments:

Reviewer 1

Comment 1: Both “obsessive-compulsive disorder” and “OCD” are listed as keywords.
Response: Revised to use only “obsessive-compulsive disorder” as the single keyword for consistency.

Comment 2: The Introduction would benefit from more mechanistic background and a clearer distinction between animal and human evidence.
Response: The Introduction has been rewritten to expand mechanistic details and now clearly differentiates established findings from animal models versus unresolved questions in human studies.

Comment 3: The meaning of “DSM and ICD criteria” should be briefly defined.
Response: Added brief definitions for DSM (Diagnostic and Statistical Manual of Mental Disorders) and ICD (International Classification of Diseases) in the Methods section.

Comment 4: Exclusion criteria should be expanded to specify whether in vitro studies, preclinical data, or psychiatric comorbidities were excluded.
Response: The Exclusion Criteria have been expanded to clarify exclusion of in vitro studies, non-validated animal/preclinical models, and subjects with major psychiatric or medical comorbidities.

Comment 5: Please clarify whether the search included all articles published in September 2025.
Response: It is now clarified that all articles published and indexed up to September 30, 2025, were included in the systematic review.

Comment 6: Section headings such as “Glutamatergic / GABAergic”, “Monoaminergic”, and “Immune & Inflammatory” would be clearer if clearly framed in the context of OCD.
Response: Section headings have been revised to explicitly highlight their OCD relevance, e.g., “Glutamatergic and GABAergic Dysfunction in OCD.”

Comment 7: Table captions and placement require revision and clarification.
Response: All tables have been provided with clear, explanatory captions as per journal requirements.

Comment 8: Claims regarding functional findings (e.g., SCFA deficiency) should be qualified to indicate small sample sizes and dietary variability.
Response: The discussion has been revised to note that SCFA findings are based on a limited number of human studies and that variability in dietary control limits the reproducibility of these results.

Comment 9: Discussion of confounding factors such as diet, medication, and comorbidities should be expanded.
Response: The impact of diet, psychotropic medication, psychiatric comorbidities, and methodological confounders on microbiome results is now discussed in depth in the revised Discussion.

Comment 10: Consider adding a schematic diagram to visually summarize the proposed gut–OCD mechanisms.
Response: A new figure has been added showing the molecular pathways linking gut microbiota to OCD pathophysiology, including SCFA deficiency, barrier disruption, inflammatory signaling, and monoaminergic alterations, to improve accessibility for all readers.

Reviewer 2 Report

Comments and Suggestions for Authors

This is a very comprehensive and relevant work, clearly demonstrating mechanistic and translational links between OCD and the microbiome.

It would be valuable to further explore the underrepresented mechanisms (eg. glutamate / GABA, HPAaxis, oxidative/nitrosative stress, neurotrophic factors)

and a visual summary figure linking biomarkers to functional microbiome changes would enhance clarity.

I recommend this manuscript for publication.

Author Response

First and foremost, I want to thank the reviewers and the prestigious journal of Life for their timely and precise process.

Here is our response to the reviewer’s comments:

Reviewer 2:

Comment: This is a very comprehensive and relevant work, clearly demonstrating mechanistic and translational links between OCD and the microbiome.

It would be valuable to further explore the underrepresented mechanisms (eg. glutamate / GABA, HPAaxis, oxidative/nitrosative stress, neurotrophic factors)

and a visual summary figure linking biomarkers to functional microbiome changes would enhance clarity.

I recommend this manuscript for publication.

Respond: My team and I profoundly appreciate your kind words. A new figure has been added showing the molecular pathways linking gut microbiota to OCD pathophysiology, including SCFA deficiency, barrier disruption, inflammatory signaling, and monoaminergic alterations, to improve accessibility for all readers.

Round 2

Reviewer 1 Report

Comments and Suggestions for Authors

Authors have revised the manuscript as per my comments.

Author Response

Comment: Authors have revised the manuscript as per my comments.

Response: My team and I appreciate your prestigious feedback.